# OpenReview forum: "BlueCodeAgent: A Blue Teaming Agent Enabled by Automated Red Teaming for CodeGen AI"
_ICLR.cc/2026/Conference — Submitted to ICLR 2026_

### Official Review · Reviewer_Vh5B · 2025-10-25

**Soundness:** 3
**Presentation:** 3
**Contribution:** 3
**Rating:** 4
**Confidence:** 3

**Summary:**

This paper addresses the underexplored challenge of blue teaming for code-generating LLMs—detecting and mitigating security risks with semantic understanding. While prior work focuses on red teaming (probing for vulnerabilities), the authors propose BlueCodeAgent, an end-to-end blue teaming agent enhanced by automated red teaming. Their framework uses red team–generated risky code examples to train a defense agent that combines constitutional reasoning, static code analysis, and agentic coordination for multi-layered protection. Evaluated on three tasks—bias detection, malicious instruction identification, and vulnerable code detection—BlueCodeAgent significantly outperforms baseline models and safety-tuned prompting. Notably, it integrates dynamic analysis in vulnerability detection to reduce false positives, a persistent issue in static-only approaches. The results show that continuous red teaming improves blue teaming by uncovering novel threats, enabling more effective, context-aware defense. This work demonstrates a practical and scalable path toward secure LLM-based code generation.

**Strengths:**

The proposed BlueCodeAgent framework is innovative in its integration of automated red teaming with multi-level defensive reasoning, combining constitutional principles, static analysis, and dynamic validation. Its demonstrated effectiveness—particularly in reducing false positives for vulnerable code detection—addresses a key practical challenge in code security. The evaluation across diverse tasks further strengthens the paper’s impact and relevance.

**Weaknesses:**

Computation Overhead: What is the computational or time cost associated with the proposed method? For blue-team applications, latency is critical since we must minimize response delays to users. The method relies on GPT-4o for constitution generation — please specify the average time per single request, without batch or multi-threaded averaging. The overhead should reflect the true per-request latency.

Online vs. Offline Constitution Generation: Why is constitution generation performed online? Please discuss the time–effectiveness trade-off between online generation (producing constitutions during testing) and offline generation (precomputing and storing constitutions in a database for retrieval based on similar inputs).

Model Choice and Scaling Behavior: Is GPT-4o strictly necessary? Describe the scaling law for constitution-generation models — for instance, how do different model sizes (e.g., 0.6B–6B parameter range) influence single-request latency and accuracy? Additionally, compare GPT-4o and GPT-5 in terms of cost, response time, and effectiveness.

**Questions:**

Please refer to the weakness.

---

> ### Author Response · Authors · 2025-11-23
> **Response to Reviewer Vh5B (Part1)**
>
> We sincerely thank you for recognizing that our work **addresses an underexplored challenge** and for highlighting the **important and practical** insight that dynamic testing can effectively reduce false positives. We also greatly appreciate your positive assessment of the **practicality and scalability** of our method, as well as your affirmation that our work is both **innovative** and promising. Below, we respond to each raised weakness and question:
>
> >Q1: Computation Overhead: What is the computational or time cost associated with the proposed method? For blue-team applications, latency is critical since we must minimize response delays to users. The method relies on GPT-4o for constitution generation — please specify the average time per single request, without batch or multi-threaded averaging. The overhead should reflect the true per-request latency.
>
> We thank you for the valuable insight. We fully agree that latency is critical in blue-teaming scenarios. We measured the average time per request when running BlueCodeAgent, and the complete results are provided in our response to Q3. Specifically, the average constitution-generation time per request using GPT-4o is 7.31 seconds, which we consider acceptable for our current setting.
>
> >Q2: Online vs. Offline Constitution Generation: Why is constitution generation performed online? Please discuss the time–effectiveness trade-off between online generation (producing constitutions during testing) and offline generation (precomputing and storing constitutions in a database for retrieval based on similar inputs).
>
> Thank you for the valuable comment. We perform online constitution generation because BlueCodeAgent receives each test case at inference time and will first compute top-k similarity to retrieve the most relevant knowledge instances before summarizing them into a constitution. Since future test cases are unknown, which is the challenging and realistic scenario we aim to address, it would be costly and impractical to precompute constitutions for all possible inputs. In terms of effectiveness, because the knowledge instances are fixed and a new test case consistently retrieves the same relevant instances, the resulting summarized constitutions are largely stable. Therefore, the impact of switching from online to offline generation on the final F1 score should be minimal.
>
> Based on your insightful suggestion, we have added a clearer discussion between online and offline generations in Section 6 (Discussion). We agree that offline constitution generation is a promising direction for reducing computational overhead and improving efficiency. In particular, offline generation can significantly reduce response latency in time-sensitive settings. A practical extension of our framework is to adopt a cache-style design, where constitutions are precomputed for knowledge clusters of representative knowledge instance groups. During testing, if the retrieved top-k knowledge matches one of the cached clusters, BlueCodeAgent can directly reuse the corresponding constitution, thereby avoiding repeated summarization.

---

> ### Author Response · Authors · 2025-11-23
> **Response to Reviewer Vh5B (Part2)**
>
> >Q3: Model Choice and Scaling Behavior: Is GPT-4o strictly necessary? Describe the scaling law for constitution-generation models — for instance, how do different model sizes (e.g., 0.6B–6B parameter range) influence single-request latency and accuracy? Additionally, compare GPT-4o and GPT-5 in terms of cost, response time, and effectiveness.
>
> Thank you for the valuable comment. In our experiment, **GPT-4o** is not the only choice. To assess how the size of the constitution generation model affects safety performance, we evaluate a range of models: `qwen3-0.6b`, `qwen3-1.7b`, `qwen3-4b`, `qwen3-8b`, `gpt4o`, and `gpt5`. We assume that `gpt4o` and `gpt5` have substantially larger parameter sizes than the Qwen models. Based on the experimental results, we do not observe a clear scaling law for constitution generation — increasing model size does not consistently improve safety outcomes. Mid-sized models such as **Qwen3-1.7B** to **8B** already achieve competitive performance comparable to `gpt4o`, while larger models like `gpt5` offer no significant additional gains. This suggests that larger summarizers are not necessarily better. Mid-sized models (1.7B–8B) and GPT-4o strike a favorable trade-off between efficiency and performance. Although GPT-5 incurs significantly higher inference time, it does not deliver significant improvements due to the overly long and verbose constitutions it produces.
>
> We have included **an updated figure summarizing all key numerical results in the revised paper (Appendix F)**; please kindly refer to it for details.
>
> Below are the statistics of different constitution generation models for constitution generation per request. Our Qwen models are run on a single NVIDIA RTX A6000 GPU:
>
> | Model       | Avg. Length per Request (chars) | Avg. Time Cost per Request (s) |
> |--|--|-|
> | Qwen3-0.6B   | 678.86  | 19.35    |
> | Qwen3-1.7B   | 1067.50  | 27.99|
> | Qwen3-4B     | 1006.48    | 32.95  |
> | Qwen3-8B     | 1190.35  | 31.45 |
> | GPT-4o       | 1619.22  | 7.31  |
> | GPT-5        | 2034.20  | 33.74 |
>
> Based on the statistics, the cost per request of GPT-4o for constitution generation is approximately 0.003\\$, and the cost per request of GPT-5 is approximately 0.004\\$.

---

### Official Review · Reviewer_FTo5 · 2025-10-25

**Soundness:** 2
**Presentation:** 3
**Contribution:** 3
**Rating:** 6
**Confidence:** 3

**Summary:**

This paper proposes BlueCodeAgent, a comprehensive blue teaming agent for code generation LLMs, which is powered by automated and diverse red teaming. The core idea is to use a pipeline where red teaming generates diverse and realistic risky instances, which are then used to create actionable constitutions guiding blue-team defenses and, in the case of code vulnerability detection, augmented with dynamic code testing. Evaluation across bias, malicious instruction, and code vulnerability detection tasks shows that BlueCodeAgent outperforms both safety prompting and recent knowledge-augmented models, achieving up to a 12.7% average F1 score improvement across several datasets.

**Strengths:**

- Integrating comprehensive automated red teaming with knowledge-enhanced blue teaming agents is an effective defense method and possesses novelty.
- This paper conducts a comprehensive evaluation of three benchmarks (bias, toxicity, and code vulnerability risks) and reports results across visible/invisible risk categories, multiple base models, and various prompt configurations, demonstrating an extensive experimental scope.

**Weaknesses:**

- BlueCodeAgent relies to some extent on the knowledge base constructed by automated red teaming, but the red teaming methods used are limited. This seems insufficient to cover all harmful categories and red teaming strategies. How does BlueCodeAgent handle cases that are not included in the knowledge base?
- BlueCodeAgent summarizes “constitutions” based on closest-matching knowledge base entries found using embedding search. However, this means blue teaming effectiveness could, in part, inherit biases or blind spots of the underlying knowledge/data, particularly if uncovered risks deviate semantically from those seen. There is little discussion or mitigation of this risk.
- The selection of baselines is relatively limited. While BlueCodeAgent demonstrates defensive capabilities, particularly in bias and malicious instruction detection, the chosen baselines are not specifically targeted defense methods. For instance, some defense strategies designed to counter jailbreaking attacks are not included. This results in BlueCodeAgent lacking a direct and meaningful comparison with such approaches.

**Questions:**

See weaknesses.

---

> ### Author Response · Authors · 2025-11-23
>
> We sincerely thank you for recognizing the **effectiveness and novelty** of our method, as well as for your positive assessment of the **comprehensiveness and thoroughness** of our evaluation. Below, we address each of the raised weaknesses and questions in detail:
>
> >Q1: BlueCodeAgent relies to some extent on the knowledge base constructed by automated red teaming, but the red teaming methods used are limited. This seems insufficient to cover all harmful categories and red teaming strategies. How does BlueCodeAgent handle cases that are not included in the knowledge base?
>
> Thank you for your valuable comment. For the harmful categories, we would like to clarify that we intentionally constrain the scope to **severe and well-defined** risk categories, enabling the red-teaming process to **systematically explore** these domains. Based on your comment, we have also added a new harmful category: **prompt injection attacks** in the updated paper in Section 4 to further broaden coverage.
>
> Regarding red-teaming strategies, as discussed in Section 3.2, our seed-based red-teaming framework is already equipped with a red-teaming agent that integrates up to **seven** tools (e.g., GCG, AmpleGCG, AutoDAN, etc.). This setup allows us to identify new combinational attack vectors, such as combining different tools to achieve stronger attacks. Future work can further extend this agent with additional red-teaming strategies and incorporate newly emerging harmful categories.
>
> We acknowledge that our current work **cannot exhaustively cover all possible harmful categories and red-teaming strategies**, as doing so would require broader community efforts and future expansion. Based on your comment, we have added a clearer discussion on the potential extension of risk categories and red-teaming strategies in Section 6 (Discussion).
>
> >Q2: BlueCodeAgent summarizes “constitutions” based on closest-matching knowledge base entries found using embedding search. However, this means blue teaming effectiveness could, in part, inherit biases or blind spots of the underlying knowledge/data, particularly if uncovered risks deviate semantically from those seen. There is little discussion or mitigation of this risk.
>
> Thank you for the constructive comment. We agree that BlueCodeAgent may inherit biases or blind spots from the underlying knowledge in certain edge cases. We have added a clearer discussion of this risk and potential mitigation strategies in Section 6. In particular, to mitigate this limitation, a potential future direction is to assign a confidence score to each constitution according to its similarity to the input. High-confidence constitutions can be relied upon more heavily, while low-confidence cases can trigger a fallback mechanism to use a general safety reminder. This approach offers a path toward improving constitution reliability and robustness in future extensions of our framework.
>
> >Q3: The selection of baselines is relatively limited. While BlueCodeAgent demonstrates defensive capabilities, particularly in bias and malicious instruction detection, the chosen baselines are not specifically targeted defense methods. For instance, some defense strategies designed to counter jailbreaking attacks are not included. This results in BlueCodeAgent lacking a direct and meaningful comparison with such approaches.
>
> Thank you for the valuable suggestions. Following your advice, we have added extensive baseline experiments. The newly included baselines are **Llama Guard**, **LlamaFirewall**, **PurpCode**, **CodeQL**, **Semgrep**, **Bandit**, and **LLM-Ensemble** variants. Detailed documentation of these baselines has been added to **Appendix Section D**. The results, as shown below, consistently demonstrate that our proposed **BlueCodeAgent significantly outperforms these baselines**. For more details, please kindly refer to Appendix D.
>
> ### Biased Code Instruction Detection
> | Method / Model | F1 |
> |---|---|
> | Llama Guard 3-8B | 0.19 |
> | LlamaFirewall | 0.00 |
> | LLM Ensemble (Initial) | 0.75 |
> | LLM Ensemble (Discussion) | 0.75 |
> | PurpCode-14b-RL | 0.58 |
> | **BlueCodeAgent (ours)** | **0.98** |
>
>
> ### Vulnerable Code Detection
> | Method / Model | F1 |
> |---|---|
> | Llama Guard 3-8B | 0.00 |
> | LlamaFirewall | 0.06 |
> | CodeQL | 0.01 |
> | Semgrep | 0.13 |
> | Bandit | 0.22 |
> | Hybrid (LLM + Tools) | 0.61 |
> | PurpCode-14b-RL | 0.67 |
> | **BlueCodeAgent (ours)** | **0.68** |
>
> ### Malicious Code Instruction Detection (RedCode-based)
> | Method / Model | F1 |
> |---|---|
> | Llama Guard 3-8B | 0.18 |
> | LlamaFirewall | 0.01 |
> | LLM Ensemble (Initial) | 0.90 |
> | LLM Ensemble (Discussion) | 0.92 |
> | PurpCode-14b-RL | 0.89 |
> | **BlueCodeAgent (ours)** | **1.00** |
>
>
> ### Malicious Code Instruction Detection (RMCbench-based)
> | Method / Model | F1 |
> |---|---|
> | Llama Guard 3-8B | 0.79 |
> | LlamaFirewall | 0.25 |
> | LLM Ensemble (Initial) | 0.90 |
> | LLM Ensemble (Discussion) | 0.90 |
> | PurpCode-14b-RL | 0.89 |
> | **BlueCodeAgent (ours)** | **0.99** |

---

> > ### Comment · Reviewer_FTo5 · 2025-11-24
> >
> > Thank you for the clarification. Although the concerns I have raised have not been entirely resolved, I acknowledge that they represent challenges that are inherently difficult to overcome. Consequently, I will be maintaining the current positive rating.

---

### Official Review · Reviewer_zFam · 2025-10-26

**Soundness:** 2
**Presentation:** 3
**Contribution:** 2
**Rating:** 4
**Confidence:** 3

**Summary:**

This paper introduces BlueCodeAgent, an end-to-end blue teaming framework for defending against security risks in code generation AI systems. The key contribution is leveraging comprehensive automated red teaming to enhance blue teaming defenses. The framework operates in two stages: (1) diverse red teaming that generates risky instances across bias instructions, malicious instructions, and vulnerable code, and (2) knowledge-enhanced blue teaming that retrieves relevant examples to generate "constitutions" (safety principles) and incorporates dynamic sandbox testing to validate vulnerabilities. Experiments across three representative tasks demonstrate improvements over baseline models and safety prompt-based defenses, with F1 scores approaching 1.0 for instruction detection tasks.

**Strengths:**

- The paper presents a novel perspective on connecting red teaming and blue teaming for code security. The idea of distilling red teaming knowledge into actionable constitutions for defense is creative. The integration of dynamic testing with LLM-based static analysis for vulnerability detection is a practical contribution that addresses the over-conservatism problem identified in prior work.
- The paper is well-structured and clearly written. Figure 2 provides a helpful overview of the framework. The distinction between principled-level defense (constitutions) and nuanced-level analysis (dynamic testing) is well-articulated.

**Weaknesses:**

- The evaluation covers only three risk categories (bias, malicious code, vulnerable code). Many other security concerns exist in code generation (e.g., privacy leaks, intellectual property violations, supply chain attacks). The "unseen risks" evaluation (Section 4) tests on different sub-categories within the same high-level risk type (e.g., different CWE types). True generalization to fundamentally different attack types remains unclear. Table 2 shows performance drops when moving from seen to unseen risks (e.g., 0.25→0.20 for bias detection), suggesting limited generalization.
- The red teaming process relies on manual enumeration of policies, bias groups, and CWE types (Section 3.2). This approach may not scale to discover novel attack vectors. The framework requires maintaining and updating knowledge bases as new vulnerabilities emerge. The paper doesn't discuss strategies for continuous learning or knowledge base maintenance.
- The safety prompt baselines are relatively weak. More sophisticated guardrail systems (e.g., Llama Guard, specialized code security models) should be compared. The LLM-ensemble baseline (Table 1) is interesting but limited to one experiment. More ensemble methods could be evaluated. There is no comparison with existing vulnerability detection tools (e.g., static analysis tools like Semgrep, CodeQL) or hybrid approaches combining LLMs with traditional security tools.

**Questions:**

1.  How does BlueCodeAgent perform on completely novel attack types not represented in the red teaming knowledge base? Can you provide evaluation on emerging threats (e.g., prompt injection attacks specific to code generation)?
2. What is the quality control process for generated constitutions? Are there cases where constitutions are incorrect, contradictory, or overly broad?

---

> ### Author Response · Authors · 2025-11-23
> **Response to Reviewer zFam (Part1)**
>
> We sincerely thank you for acknowledging BlueCodeAgent’s **novelty** in connecting red teaming and blue teaming for code security, as well as the **creativity** of distilling red-teaming knowledge into actionable constitutions for defense. We also appreciate the recognition that our integration of dynamic testing with LLM-based static analysis is a **practical contribution** to reduce false positives (FP). We are also grateful for your praise that our paper is **well-written**. Below, we respond to each weakness and question:
>
> >W1.1: The evaluation covers only three risk categories (bias, malicious code, vulnerable code). Many other security concerns exist in code generation (e.g., privacy leaks, intellectual property violations, supply chain attacks).
>
> Thanks for the valuable comment. We choose these three categories because these are the most **representative**, **widely studied**, and **severe** risks in prior work. Following your suggestion, we additionally include experiments on a new risk category—**prompt injection attacks** in the code generation domain. The results are updated in the paper, and we also explain the results of the prompt injection attack later in response to your question Q1.
>
> We would like to kindly clarify that our main contribution lies in proposing an end-to-end framework that enables automatic red teaming and knowledge-enhanced blue teaming, along with extensive experimental insights. We acknowledge that expanding to additional risk categories would require substantial effort in designing new red-teaming strategies and constructing corresponding benchmarks. However, future work can leverage our automatic red-teaming framework to cover a broader range of attacks and integrate novel strategies to address new types of risks. This detailed discussion about the extension of risk categories and red-teaming strategies is also included in our updated section 6 in the PDF.
>
> >W1.2: The "unseen risks" evaluation (Section 4) tests on different sub-categories within the same high-level risk type (e.g., different CWE types).
>
> Thanks for the valuable comment. We would like to clarify that although our “unseen risks” evaluation tests on different sub-categories within the same high-level risk type (e.g., different CWE types), this setting already reflects cross-type generalization within vulnerable code categories. Prior studies (e.g., SVEN [1], PrimeVul[2]) typically train and test on the overlapped CWE types, while our evaluation demonstrates the ability of BlueCodeAgent to generalize across different sub-categories, such as CWEs, that were not seen in the knowledge.
>
> [1] Large language models for code: Security hardening and adversarial testing  Jingxuan He, Martin Vechev ACM CCS 2023
> [2] Vulnerability Detection with Code Language Models: How Far Are We? Yangruibo Ding et al. ICSE 2025
>
> > W1.3: True generalization to fundamentally different attack types remains unclear. Table 2 shows performance drops when moving from seen to unseen risks (e.g., 0.25→0.20 for bias detection), suggesting limited generalization.
>
> Thank you for the insightful comment. We acknowledge that performance inevitably decreases when the model encounters different categories in the knowledge and test, and that’s also our main purpose to conduct the experiments and test the generalization of the pipeline. While generalization is a fundamental challenge in machine learning, we observe that our pipeline has reasonably high generalization capability as the method has leveraged the general knowledge/constitution rather than only instance-based training. BlueCodeAgent is equipped with end-to-end automatic red-teaming and knowledge-enhanced blue-teaming to strengthen generalization. As shown in Section 4, our results already demonstrate improved generalization, since we evaluate on out-of-distribution sub-categories and still observe consistent performance gains. We thank the reviewer again for raising this question. We will further clarify and elaborate on the generalization problem in the revised version.

---

> ### Author Response · Authors · 2025-11-23
> **Response to Reviewer zFam (Part2)**
>
> > W2.1: The red teaming process relies on manual enumeration of policies, bias groups, and CWE types (Section 3.2). This approach may not scale to discover novel attack vectors.
>
> Thanks for the valuable comment. Discovering fundamentally novel attack vectors is interesting, and the red teaming exploration has the potential, but we haven’t focused on this perspective in this work yet. On the red-teaming side, our framework implements an **end-to-end automatic red-teaming pipeline** designed to cover comprehensive and representative known severe risk categories. We intentionally constrain the scope to severe risk categories to ensure that the BlueCodeAgent is able to detect these risks, which are critical, based on the systematic and scalable risk uncovery process of the red teaming.
>
> As for the novel attack vectors, as discussed in Section 3.2, our seed-based red-teaming is equipped a red-teaming agent with up to seven tools (e.g., GCG, AmpleGCG, AutoDAN, etc.), through which we have identified **new combinational attack vectors**—for example, combining different tools to achieve successful attacks. Future work can further extend this red-teaming agent with additional strategies to support the discovery of novel attack vectors.
>
> Due to space and scope limitations, we cannot include all potential red-teaming extensions in this paper. Following your suggestions, we have included a clearer discussion of these limitations in the revised version.
>
> > W2.2: The framework requires maintaining and updating knowledge bases as new vulnerabilities emerge. The paper doesn't discuss strategies for continuous learning or knowledge base maintenance.
>
> Thanks for the suggestions. We would like to clarify that maintaining and updating knowledge bases is an **automatic process** in our framework. Our manual enumeration of policies, bias groups, and CWE types knowledge is already a comparatively **lightweight effort** in red-teaming. And through automatic red teaming, the generated knowledge data can continuously be added to the knowledge base. As for new attack vectors and vulnerabilities, future users can add new red-teaming tools into our red-teaming process and update new knowledge automatically.
> Following your suggestion, we also added a clearer discussion about continuous learning in our updated section 6 (Discussion).
>
> > W3: The safety prompt baselines are relatively weak. More sophisticated guardrail systems (e.g., Llama Guard, specialized code security models) should be compared. The LLM-ensemble baseline (Table 1) is interesting but limited to one experiment. More ensemble methods could be evaluated. There is no comparison with existing vulnerability detection tools (e.g., static analysis tools like Semgrep, CodeQL) or hybrid approaches combining LLMs with traditional security tools.
>
> Thank you for the valuable suggestions. Following your advice, we have added extensive baseline experiments. The newly included baselines are **Llama Guard**, **LlamaFirewall**, **PurpCode**, **CodeQL**, **Semgrep**, **Bandit**, and **LLM-Ensemble** variants. Detailed documentation of these baselines has been added to **Appendix Section D**. The results, as shown below, consistently demonstrate that **our proposed BlueCodeAgent significantly outperforms these baselines**. For detailed settings, please kindly refer to Appendix D.
>
> ### Biased Code Instruction Detection
> | Method / Model | F1 |
> |---|---|
> | Llama Guard 3-8B | 0.19 |
> | LlamaFirewall | 0.00 |
> | LLM Ensemble (Initial) | 0.75 |
> | LLM Ensemble (Discussion) | 0.75 |
> | PurpCode-14b-RL | 0.58 |
> | **BlueCodeAgent (ours)** | **0.98** |
>
>
> ### Vulnerable Code Detection
> | Method / Model | F1 |
> |---|---|
> | Llama Guard 3-8B | 0.00 |
> | LlamaFirewall | 0.06 |
> | CodeQL | 0.01 |
> | Semgrep | 0.13 |
> | Bandit | 0.22 |
> | Hybrid (LLM + Tools) | 0.61 |
> | PurpCode-14b-RL | 0.67 |
> | **BlueCodeAgent (ours)** | **0.68** |
>
>
> ### Malicious Code Instruction Detection (RedCode-based)
> | Method / Model | F1 |
> |---|---|
> | Llama Guard 3-8B | 0.18 |
> | LlamaFirewall | 0.01 |
> | LLM Ensemble (Initial) | 0.90 |
> | LLM Ensemble (Discussion) | 0.92 |
> | PurpCode-14b-RL | 0.89 |
> | **BlueCodeAgent (ours)** | **1.00** |
>
>
> ### Malicious Code Instruction Detection (RMCbench-based)
> | Method / Model | F1 |
> |---|---|
> | Llama Guard 3-8B | 0.79 |
> | LlamaFirewall | 0.25 |
> | LLM Ensemble (Initial) | 0.90 |
> | LLM Ensemble (Discussion) | 0.90 |
> | PurpCode-14b-RL | 0.89 |
> | **BlueCodeAgent (ours)** | **0.99** |

---

> ### Author Response · Authors · 2025-11-23
> **Response to Reviewer zFam (Part3)**
>
> >Q1: How does BlueCodeAgent perform on completely novel attack types not represented in the red teaming knowledge base? Can you provide evaluation on emerging threats (e.g., prompt injection attacks specific to code generation)?
>
> Thanks for the comment. Following your advice, we additionally include experiments on a new risk category—**prompt injection attacks** in the code generation domain.
>
> We did two separate experiments on **prompt injection attacks**. (1) We use existing knowledge from bias and malicious code tasks to evaluate new prompt injection attacks. The malicious components in these prompt injection tasks remain the same as in the biased and malicious code instruction categories. The key difference is that prompt injection semantics—such as “ignore all the previous instructions”—are embedded within the test cases. Following prior works [1], we construct our test cases using **five representative prompt injection attack categories**: naive attack, escape characters, context ignoring, fake completion, and combined attack. We test the same base LLMs settings as in Section 4. The results are shown in the table below, **BlueCodeAgent effectively leverages existing knowledge to identify unsafe tasks with prompt injection attacks.**
>
> ### F1 score on bias code instruction with prompt injection attack
> | **Method**                  | **Llama** | **Qwen** | **GPT-4o** |
> |-----------------------------|-----------|-----------|------------|
> | Directly testing            | 0.73      | 0.57      | 0.71       |
> | General safety reminder     | 0.63      | 0.50      | 0.68       |
> | Fine-grained safety reminder| 0.79      | 0.57      | 0.89       |
> | **BlueCodeAgent**           | **0.86**  | **0.70**  | **0.98**   |
>
> ### F1 score on malicious code instruction with prompt injection attack (RedCode-based)
> | **Method**                  | **Llama** | **Qwen** | **GPT-4o** |
> |-----------------------------|-----------|-----------|------------|
> | Directly testing            | 0.72      | 0.88      | 0.91       |
> | General safety reminder     | 0.75      | 0.88      | 0.90       |
> | Fine-grained safety reminder| 0.80      | 0.88      | 0.94       |
> | **BlueCodeAgent**           | **0.97**  | **1.00**  | **0.99**   |
>
> ### F1 score on malicious code instruction with prompt injection attack (RMCbench-based)
> | **Method**                  | **Llama** | **Qwen** | **GPT-4o** |
> |-----------------------------|-----------|-----------|------------|
> | Directly testing            | 0.49      | 0.85      | 0.94       |
> | General safety reminder     | 0.53      | 0.81      | 0.87       |
> | Fine-grained safety reminder| 0.58      | 0.89      | 0.94       |
> | **BlueCodeAgent**           | **0.93**  | **1.00**  | **1.00**   |
>
> (2) Furthermore, we design and complement the full BlueCodeAgent pipeline from red-teaming to blue-teaming for the prompt injection risk category. The overall F1 score results are shown in the table below. BlueCodeAgent consistently achieves stronger or near-perfect performance across models. For more details, please refer to our updated paper in Section 3, Section 4, and Figure 3.
>
> | **Method**              | **Llama** | **Qwen** | **GPT-4o** |
> |----------------------------------|-----------|-----------|------------|
> | **Directly testing**             | 0.14      | 0.95      | 0.83       |
> | **General safety reminder**      | 0.22      | 0.97      | 0.86       |
> | **Fine-grained safety reminder** | 0.34      | 0.99      | 0.96       |
> | **BlueCodeAgent**                | **0.92**  | **0.99**  | **0.98**   |
>
> [1] Formalizing and Benchmarking Prompt Injection Attacks and Defenses,  Yupei Liu et al. USENIX Security Symposium, 2024
>
> >Q2: What is the quality control process for generated constitutions? Are there cases where constitutions are incorrect, contradictory, or overly broad?
>
> We thank you for the detailed observation. Since our constitutions are derived from our strict red-teaming data generation pipeline, each generated datapoint is grounded in an aspect of the corresponding risk category, so each constitution is grounded in an aspect of the corresponding risk category. So they are highly reliable and unlikely to be incorrect and contradictory. We agree that constitutions may sometimes be overly broad, but this reflects an inherent trade-off: if constitutions are too specific, they may lead to overfitting in blue teaming and reduce generalization, whereas overly broad constitutions may fail to capture important unsafe patterns. Based on your comment, we have added a dedicated discussion on constitution reliability in Section 6 (Discussion).

---

### Official Review · Reviewer_GkSu · 2025-11-02

**Soundness:** 3
**Presentation:** 3
**Contribution:** 3
**Rating:** 6
**Confidence:** 3

**Summary:**

The paper introduces BlueCodeAgent, a framework for "blue teaming" against code generation LLMs. BlueCodeAgent tackles security challenges like detecting biased instructions by integrating knowledge derived from an automated "red teaming" pipeline. This system leverages the offensive data to generate safety rules that guide the defensive LLM agent and employs dynamic testing to verify potential vulnerabilities and reduce false positives. Evaluation across multiple benchmarks demonstrates that this knowledge-enhanced, agent-based approach significantly improves risk detection and mitigation compared to baseline and simple safety-prompt defences.

**Strengths:**

- BlueCodeAgent achieves significant gains over the baseline models and safety prompt-based defenses, demonstrating much more effective and context-aware risk detection and mitigation. It consistently performs well on both seen and unseen risks
- Red-teaming can empower effective blue-teaming defenses, showing that red teaming benefits blue teaming by continuously identifying new vulnerabilities

**Weaknesses:**

- The proposed methods are mostly based on prompt engineering and the technical contribution is therefore limited for this venue
- The definition of blue teaming is not presented in the paper. It is only clear from the context, but I would recommend to add a clear defintion early in the paper to show the contribution
- Limited Scope of Risk Categories: The current evaluation focuses on three representative code-related tasks: bias instruction detection, malicious instruction detection, and vulnerable code detection
- To enhance its real-world utility, BlueCodeAgent needs to be scaled up to operate at the file and repository levels. This scaling effort would require the agent to be equipped with more advanced context retrieval tools and memory components

**Questions:**

- How are vulnerabilities detected?

---

> ### Author Response · Authors · 2025-11-23
>
> We sincerely thank you for acknowledging BlueCodeAgent’s **effectiveness**, including its significant gains over the baselines and its **generalizability**—consistently performing well on both seen and unseen risks. Below, we respond to each raised weakness and question:
>
> >W1: The proposed methods are mostly based on prompt engineering and the technical contribution is therefore limited for this venue
>
> Thanks for the valuable comment. We would like to clarify and emphasize that our main contributions include:
>
> (1) **A diverse red-teaming pipeline**, encompassing policy-based instance generation, seed-based adversarial prompt optimization, and knowledge-driven vulnerability generation.
>
> (2) **Knowledge-enhanced blue teaming**, introducing a novel and effective approach where constitution generation further improves the blue-teaming process.
>
> (3) **Principled-level defense and nuanced-level analysis**, integrating constitution-based detection with sandbox-based code execution inspection in the vulnerable code detection domain, supported by ablation insights and substantial technical effort.
>
> (4) **Extensive experimental findings**, demonstrating the effectiveness of knowledge, the complementary benefits of constitutions and dynamic testing, and generalizability across both seen and unseen risk scenarios.
>
> (5) Following the reviewers’ suggestions, we have also added additional **comprehensive experiments** during rebuttal, as mentioned in the global response. These additional results provide further insights that can benefit future knowledge-enabled frameworks.
>
> > W2: The definition of blue teaming is not presented in the paper. It is only clear from the context, but I would recommend to add a clear defintion early in the paper to show the contribution
>
> Thanks for the valuable comment. Following your suggestion, we have **added a clearer definition** of blue teaming in the updated paper. As introduced in Section 3.1 Overview, our blue teaming task is formulated as a risky instance detection problem, such as identifying biased and malicious code instructions and vulnerable code, where the goal of the models or agents is to determine whether an input test case is risky or not. BlueCodeAgent aims to effectively distinguish risky instances from benign ones, thereby building clearer decision boundaries for code-generation security tasks, which is critical as coding assistants are widely applied currently.
>
> >W3: Limited Scope of Risk Categories: The current evaluation focuses on three representative code-related tasks: bias instruction detection, malicious instruction detection, and vulnerable code detection
>
> Thanks for the valuable comment. We choose these three categories because these are the most **representative**, **widely studied**, and **severe** risks in prior work. Following your suggestion, we additionally include experiments on a new risk category—**prompt injection attacks** in the code generation domain. We design and implement the full BlueCodeAgent pipeline from red-teaming to blue-teaming for the prompt injection risk category. We test the same base LLMs settings as in Section 4. The overall F1 score results are shown in the table below. BlueCodeAgent consistently achieves stronger or near-perfect performance across models. For more details, please refer to our updated paper in Section 3, Section 4, and Figure 3.
>
> | **Method**   | **Llama** | **Qwen** | **GPT-4o** |
> |--|-----|---|-----|
> | **Directly testing**  | 0.14      | 0.95      | 0.83       |
> | **General safety reminder**  | 0.22      | 0.97      | 0.86       |
> | **Fine-grained safety reminder** | 0.34      | 0.99      | 0.96       |
> | **BlueCodeAgent**              | **0.92**  | **0.99**  | **0.98**   |
>
> >W4: To enhance its real-world utility, BlueCodeAgent needs to be scaled up to operate at the file and repository levels. This scaling effort would require the agent to be equipped with more advanced context retrieval tools and memory components
>
> Thanks for the valuable comment. We agree that scaling BlueCodeAgent to the repository level requires equipping the agent with more advanced context retrieval tools and memory components. This direction will be an important improvement in future work. We will add the related discussion in our revision.
>
> >Q1: How are vulnerabilities detected?
>
> Thanks for the valuable comment. For BlueCodeAgent,  vulnerabilities are detected through **constitution-guided static analysis** and **dynamic sandbox testing**. BlueCodeAgent first analyzes potential vulnerable code using security constitutions, and if risks are suspected, based on the reasoning, a reliable test case generation model (Claude) will generate test code to verify the hypothesis and execute test cases in an isolated sandbox environment to verify real vulnerabilities. Finally, BlueCodeAgent combines static analysis, test code, execution result, and constitution for final judgment. We have made this part clear in our updated paper in Section 3.3.

---

### Author Response · Authors · 2025-11-23
**Global Response**

We sincerely thank all the reviewers for their insightful, constructive, and valuable comments and suggestions. We are delighted that the reviewers recognize our method as **novel**, **creative**, **innovative**, **effective**, **generalizable**, **scalable**, and **practical**. We also greatly appreciate the acknowledgement that the paper is **well-written**, the experiments are **comprehensive** and **extensive**, and that our work addresses an **under-explored challenge**  and represents an important direction.

Below, we summarize our major updates made during the rebuttal:

1. **Extensive Baselines.**
We added comprehensive baseline experiments showing that our method consistently outperforms strong baselines, including **Llama Guard**, **LlamaFirewall**, **PurpCode**, **CodeQL**, **Semgrep**, **Bandit**, and **LLM-Ensemble** variants in Appendix D (following the suggestions of Reviewer zFam and Reviewer FTo5).

2. **New Risk Category: Prompt Injection.**
We added experiments evaluating whether BlueCodeAgent can detect new prompt-injection attacks using existing knowledge in Appendix E. In addition, we introduced **prompt injection** as a new risk category throughout the entire BlueCodeAgent pipeline in Section 4 (following the suggestions of Reviewer GkSu, Reviewer zFam, and Reviewer FTo5).

3. **Scaling Law Analysis for Constitution Generation Models.**
We added experiments and analysis of the constitution generation model scaling in Appendix F, showing that mid-sized models already achieve good performance (following the suggestions of Reviewer Vh5B).

4. **Enhanced Discussion Section.**
We added new discussions on (i) **Extension of risk categories and red-teaming strategies for continuous learning**, (ii) **Reliability of constitutions**, and (iii) **Online and offline constitution generation** in Section 6 following the suggestions of Reviewer GkSu, Reviewer zFam, and Reviewer FTo5.

All updated components are highlighted in **maroon** in the revised PDF. Please refer to the updated version for more details.

---

### Author Response · Authors · 2025-12-03
**Rebuttal Summary**

Dear Reviewers and Area Chairs,

We sincerely thank all reviewers and area chairs for their time and thoughtful feedback on our submission. We are glad that the reviews commonly recognize the **novelty** of our work (Reviewer zFam, Reviewer FTo5, Reviewer Vh5B), as well as the **effectiveness** and **comprehensive evaluation** of our end-to-end framework for automatic red teaming and knowledge-enhanced blue teaming.

We have carefully addressed every concern in our point-by-point responses and incorporated the corresponding revisions into the updated manuscript. Below, we summarize the additional experiments, key clarifications, and new analyses included in our rebuttal:

1. **New Risk Category: Prompt Injection**
Following the reviewers’ suggestions on risk categories analysis (Reviewer GkSu, Reviewer zFam, and Reviewer FTo5), we added prompt injection in code generation as a new risk category throughout the entire BlueCodeAgent pipeline (Section 4). We also conducted new experiments (Appendix E) evaluating whether BlueCodeAgent can detect new prompt-injection attacks using only existing knowledge. Results confirm BlueCodeAgent’s superiority over baselines.

2. **Clarification on Generalization**
Following Reviewer zFam’s suggestion about generalization, we added additional discussion to analyze and show that our method does in fact demonstrate strong generalization capability. While generalization is a fundamental challenge in machine learning, we observe that our pipeline has reasonably high generalization capability as the method has leveraged the general knowledge rather than only instance-based training. Although our “unseen risks” evaluation tests on different sub-categories within the same high-level risk type (e.g., different CWE types), this setting already reflects cross-type generalization, especially considering that prior work typically trains and tests on the same sub-categories.  As shown in Section 4, our results already demonstrate improved generalization, since we evaluate on out-of-distribution sub-categories and still observe consistent performance gains.

3. **Extensive Baselines**
Following the reviewers' questions about baselines (Reviewer zFam and Reviewer FTo5), we conducted additional comprehensive baseline experiments showing that our method consistently outperforms strong baselines, including Llama Guard, LlamaFirewall, PurpCode, CodeQL, Semgrep, Bandit, and LLM-Ensemble variants in Appendix D

4. **More Discussion on Potential Extension**
Following the suggestions of Reviewers GkSu, zFam, and FTo5, we expanded the discussion section: the extension of risk categories, red-teaming strategies for continuous learning, the reliability of constitutions, and online vs. offline constitution generation. These additions outline potential directions and practical methods for further extending our work.

5. **Scaling Law Analysis for Constitution Generation Models**
To answer Reviewer Vh5B’s question regarding the choice of constitution-generation models, we added new experiments and analyses on model scaling in Appendix F. We evaluate a range of Qwen models, GPT-4o, and GPT-5, and compare their performance. We find that mid-sized models already achieve strong performance, and we do not observe a clear scaling law for constitution-generation models. By examining the length of the generated constitutions, we also find that overly short constitutions fail to capture important unsafe patterns, while overly long or verbose constitutions tend to overfit during blue teaming and reduce generalization.

We thank all the reviewers and area chairs again for their efforts in helping us improve BlueCodeAgent. We hope our responses fully address the raised concerns. Given the reviewers’ encouraging and constructive assessments, as well as our substantial efforts in revising the work, we believe the paper has been significantly strengthened.

Best regards,
Authors

---

### Meta-Review · Area_Chair_QNgQ · 2026-01-07

**Summary:**

Major concerns include:
1. Limited scope of the work: Several reviewers mentioned the limited risk categories covered by the proposed approach and dataset as a main concern. There are also concerns about the limited "unseen task" evaluation and the function-level (as opposed to file- or repository-level) code generation setting;
2. Limited generalizability of the proposed approach: Reviewers are concerned that the proposed agent relies on manual enumeration of policies, bias groups, and CWE types and thus cannot generalize beyond the pre-defined scope (i.e., the seen risk categories);
3. Baselines should be stronger.

**Reviewer Concerns:**

Authors responded to all the major concerns:
1. Authors justified that the risk categories are carefully selected. To expand the scope, they added one risk category (i.e., prompt injection). The unseen task setting follows the design of prior work. File or repository-level code generation can be future work.
2. Authors justified that they focus on the end-to-end framework design for known risk categories.
3. New baselines were added; the proposed approach outperformed them.

**Reviewer Scores:**

One reviewer (FTo5) responded that they would keep the original score (6). It is a little difficult to judge if reviewers who are concerned about the limited scope of the work would increase their scores. The reason is that the proposed approach is designed to target only known risk categories. Generalization to unseen risk categories is not a focus of this work. Given that, reviewers may or may not think their concern is well addressed.

---

### Decision · Program_Chairs · 2026-01-26

Reject